# The Feasibility of the Diabetes Self-Management Coaching Program in Primary Care: A Mixed-Methods Randomized Controlled Feasibility Trial

**DOI:** 10.3390/ijerph21081032

**Published:** 2024-08-06

**Authors:** Fikadu Ambaw Yehualashet, Dorothy Kessler, Segenet M. Bizuneh, Catherine Donnelly

**Affiliations:** 1Department of Rehabilitation Science, School of Rehabilitation Therapy, Queen’s University, 31 George St., Kingston, ON K7L 3N6, Canada; dk75@queensu.ca (D.K.); catherine.donnelly@queensu.ca (C.D.); 2Department of Nursing, School of Nursing, College of Medicine and Health Science, University of Gondar, Gondar P.O. Box 196, Ethiopia; 3Department of Internal Medicine, School of Medicine, College of Medicine and Health Science, University of Gondar, Gondar P.O. Box 196, Ethiopia; segenetnew@gmail.com

**Keywords:** feasibility, diabetes, self-management, acceptability, fidelity, coaching, Ethiopia

## Abstract

Background: Diabetes mellitus, a chronic metabolic disorder associated with elevated blood sugar levels, is a significant cause of morbidity, mortality, and disability globally. The rampant rise in the prevalence of diabetes presents a public health burden and a challenge to the primary care setting. Diabetes self-management coaching is an emergent, client-centered, solution-focused approach to enhance self-efficacy and self-care behavior, control glycemia, and prevent acute and chronic complications. Currently, there is no diabetes self-management support strategy in the primary care setting in Ethiopia. Therefore, this study assessed the feasibility, acceptability, and fidelity of implementing the Diabetes Self-Management Coaching Program in primary care in Ethiopia. Method: A single-center, single-blinded, parallel group mixed-methods feasibility randomized control design was applied to assess the feasibility, acceptability, and fidelity of the Diabetes Self-Management Coaching Program in primary care. Adult patients with type 2 diabetes with HbA1c ≥ 7%, taking diabetic medication, and living in Gondar town were included in the study. A block randomization technique with a block size of four was used to allocate participants into the treatment and control groups. The treatment group attended a 12-week Diabetes Self-Management Coaching Program in addition to undergoing usual care, while the control group received the usual care for the same period. Data were collected at baseline, at the end of the intervention, and after the follow-up period. Descriptive statistics such as the frequency, mean, median, and standard deviations were computed. Based on the normality assessment, the baseline group difference was examined using the independent sample Student’s *t*-test, the Mann–Whitney U test, and the chi-square test. Result: This study’s eligibility, recruitment, retention, and adherence rates were 23%, 70%, 90%, and 85%, respectively. Both the qualitative and quantitative findings show that the program was feasible to implement in primary care and acceptable to the participants. The fidelity assessment of the Diabetes Self-Management Coaching Program indicates an appropriate intervention implementation. Conclusions: This study demonstrated remarkable recruitment, retention, and adherence rates. The Diabetes Self-Management Coaching Program was feasible, acceptable, and implementable in primary care in Ethiopia. As a result, we recommend that a large-scale multi-center cluster randomized controlled trial with an adequate sample can be designed to evaluate the effect of the DSM Coaching Program on clinical and behavioral outcomes.

## 1. Background

Diabetes mellitus has become a significant global public health challenge with an alarmingly mounting prevalence of morbidity, mortality, and disability despite advancements in diagnostic and therapeutic technologies. According to the 2021 International Diabetes Federation report, globally, more than 536.6 million people are living with diabetes, and this number has been projected to reach 783.2 million by 2045 [1]. The majority of individuals with diabetes (80%) live in low- and middle-income countries (LMICs) [2,3] where the health systems have limited budgets, poor infrastructure, and inadequate human resources. The 2021 International Diabetes Federation report indicates a 3.2% prevalence of diabetes in Ethiopia, with over 57% undiagnosed cases of diabetes [4]. Another systematic review and meta-analysis study in 2021 shows a 6.5% pooled prevalence of diabetes in Ethiopia [5].

Self-management support strategies are recognized as an essential component of chronic disease management and secondary prevention [6]. Various self-management strategies have been tailored to different chronic disease conditions, including diabetes. Diabetes self-management is the ability of an individual to perform self-care activities regularly [7]. These self-care activities include diet management, regular physical exercise, blood glucose monitoring, foot care, and medication compliance [8]. Evidence shows that engaging people in self-management is paramount in reducing hospital visits, minimizing hospitalization rates, and improving health status and illness management [9]. Health coaching, a self-management support strategy, shows promise in improving the clinical, personal, and behavioral outcomes of people living with diabetes [10,11,12,13]. Unlike routine diabetes health education programs, diabetes health coaching is a client-centered, solution-focused, and goal-directed approach to empowering the self-management ability of individuals with diabetes [14].

With an alarmingly rising burden of chronic health conditions, primary care settings play a pivotal role in providing self-management support for individuals with diabetes [15,16]. Primary care in Ethiopia focuses on health promotion, disease prevention, acute and chronic infectious disease management, and maternal and child health [17]. The role of self-management in promoting health and preventing complications and premature deaths among individuals with chronic health conditions has been overlooked in the primary healthcare system in Ethiopia. Given the country’s growing number of people with chronic health conditions, and as a first point of access, primary care in Ethiopia is well placed to integrate diabetes self-management support. Leveraging the robust health extension program allows nurses and/or health extension workers to visit individual households and provide health education to individuals with type 2 diabetes. While the Ethiopian Chronic Disease and Injury Management Guideline recommends diabetes care for individuals with type 2 diabetes in primary care [18] diabetes self-management support services are not integrated or unavailable in primary care settings. On top of this, a culturally tailored and effective diabetes self-management strategy is required to improve the quality of life of individuals with type 2 diabetes in this setting. Health systems in Ethiopia provide traditional diabetes education for individuals with type 2 diabetes, which is content-focused and not patient-centered or goal-oriented; hence, it is less effective in changing patients’ behaviors [19]. Therefore, the DSM Coaching Program, which shows promising health outcomes and reduces diabetes-related hospital admissions and medical costs in different countries, could be an effective approach in the primary care setting in Ethiopia. Therefore, the current study examined the feasibility, acceptability, and fidelity of the DSM Coaching Program in primary care in Ethiopia.

Objectives: This study’s overarching goal is to determine the feasibility, acceptability, and fidelity of implementing the DSM Coaching Program among individuals with type 2 diabetes in primary care in Ethiopia.

The specific objectives are as follows:⮚Determine the feasibility of implementing the DSM Coaching Program.⮚Assess the acceptability of the DSM Coaching Program by individuals with type 2 diabetes.⮚Determine the fidelity of implementing the DSM Coaching Program.

## 2. Method and Materials

Study Design: This study used a concurrent mixed-methods, single-blind feasibility parallel group randomized controlled trial (RCT) design to assess the feasibility and acceptability of the DSM Coaching Program.

Study Setting: This study was conducted from 1 November 2022 to 30 April 2023 in the primary care setting in Ethiopia. The health system in Ethiopia is a three-tier system with primary-, secondary-, and tertiary-level care. Primary care is the gateway and the closest level of care for the community, and services are delivered in a primary care unit. The primary care unit encompasses a district primary hospital, a health center, and five health posts. This study was conducted in primary care settings of Gondar City, Amhara region, Ethiopia. Although this study was conducted in primary care settings, we recruited individuals with type 2 diabetes from the University of Gondar Hospital chronic disease outpatient clinic, hereafter named the “diabetic clinic”, as this was the location where most patients received routine diabetic care and offered the most robust location for recruitment.

Gondar City has one referral hospital, ten primary care centers, and more than thirty private health facilities [20]. The hospital serves more than 2600 individuals with diabetes in the diabetic clinic, of which more than 1300 patients have type 2 diabetes [21]. In addition, because of an ineffectual referral system in the city, the hospital is serving as a primary, secondary, and tertiary care center. The intervention was carried out in the selected primary care settings (health centers) and participants’ homes. Participants chose two primary care locations among four health centers. As a result, the Azezo and Gondar City health centers were chosen for the group coaching.

### 2.1. Participant Eligibility

This study included individuals who had been receiving diabetic care at the University of Gondar Comprehensive Specialized Hospital for at least six months, were living in Gondar City, were taking anti-diabetic medications, had a recent HbA1c ≥ 7% (within three months), and were between 18 and 65 years old. However, this study excluded individuals with clinically confirmed mental illness, pregnant mothers [22], and individuals with chronic diabetes complications, including neuropathy, nephropathy, retinopathy, stroke, gangrene, and cardiovascular disease. This study also excluded individuals who were seriously ill and hospitalized in the past three months. In addition, we excluded people with visual and hearing impairments due to a lack of assistive devices and translators to accommodate their needs. Furthermore, as the DSM Coaching Program requires participants to be physically active and exercise regularly, individuals with lower extremity amputation or palsy were excluded from the study.

Recruitment: Participant recruitment was conducted from 15 September 2022 to 30 October 2022 in the University of Gondar comprehensive referral hospital. The hospital has been serving as a primary, secondary, and tertiary care center because of the unstructured referral system in the city.

Audio and visual recruitment information that contained the recruitment information and process of enrollment were prepared in Amharic, the local language. We used a tape recorder to play the audio recording in the waiting area of the diabetic clinic. In addition, a poster that contained the recruitment information and the process of enrollment was prepared in a local language and was displayed in the waiting area.

Nurses working in the diabetes clinic received two hours of orientation training on the screening criteria and process of recruitment, and they screened potential participants using a structured screening tool. Individuals with type 2 diabetes who passed the screening process were contacted by the research assistant. The research assistant discussed the informed consent form with the participants and offered to sign it. Baseline data were collected from consented participants and randomly allocated to the study groups.

### 2.2. Sample Size Determination

The goal of the current feasibility study was to provide sufficient information for the plausibility and applicability of a powered definitive study. There is no single guideline for calculating the sample size for pilot and feasibility studies [23,24]. However, researchers used different approaches to estimate the sample size for pilot/feasibility trials, such as the size of the definitive trial, the estimated effect size, the type of outcome variable, or flat rates, as the rule of thumb [25]. The current study followed Hertzog’s suggestion of sample size for pilot studies [26]. Accordingly, we recruited 40 individuals with uncontrolled type 2 diabetes as a sample size and assigned 20 participants in each arm.

### 2.3. Randomization and Blinding

Randomization and allocation: This study used a block randomization technique with a block size of four [27] to assign participants to the treatment and control groups. An external researcher with no other role in the study manually generated the random allocation sequence [28]. From a block of four denoted by “T” for treatment and “C” for control, six possible combinations were generated: TTCC, TCTC, TCCT, CCTT, CTTC, and CTCT. These combinations were written on a piece of paper, rolled for a lottery, and placed in a bowl. Each combination had a chance to be selected ten times by rolling it back and adding it to the bowl. A nurse working in another department drew the lottery ten times until it reached 40. The external researcher generated the random allocation sequence based on the order of the letters (T and C) in each piece of paper selected by lottery [28]. To conceal the allocation of the group, the external researcher prepared an opaque sealed envelope that contained group allocation information [28,29]. These envelopes were sequentially numbered based on the pattern of block randomization. The research assistant received the sealed envelopes from the external researcher and randomly allocated eligible and consented participants by opening the envelope based on their enrollment sequence.

Blinding: Because of the nature of the intervention, it was not easy to mask the study participants and the interventionist. However, the assessors did not know the group allocations of the participants. The research assistant provided a five-minute orientation for each participant just before data collection to prevent potential disclosure of the group allocation. To ensure blinding, we used one data collector for each assessment period.

Intervention arms of the study: This study had two arms: the control group, who continued receiving the usual care in the diabetic clinic, and the treatment group, who attended a 12-week DSM Coaching Program and received the usual care. The DSM Coaching Program had six group-based and four individual home-based coaching sessions. A group of five to six participants attended six biweekly group coaching sessions, with each session lasting two hours. These sessions encompassed an introduction and the basics of diabetes, goal setting, diet management, physical exercise, glucose monitoring and medication compliance, and foot care. The treatment group also attended four individual home-based coaching sessions for 30 min, which addressed dietary management, regular exercise, blood glucose monitoring and medication compliance, and foot care.

### 2.4. Data Collection

This study utilized qualitative and quantitative data sources to examine the feasibility, acceptability, and fidelity of the DSM Coaching Program. We computed outcome data from screening reports, session attendance, and intervention records. In addition, we used the Treatment Acceptability/Adherence Scale (TAAS) to assess the acceptability of the DSM Coaching Program among the study participants who attended the DSM Coaching Program. An experienced qualitative researcher conducted the in-depth interviews using a semi-structured interview guide prepared by the research team.

### 2.5. The Feasibility Outcomes of the Study

The current study addressed the eligibility rate, recruitment rate, adherence rate, retention rates, and acceptability of the DSM Coaching Program. The predefined progression criteria determine the overall feasibility of the DSM Coaching Program.

Eligibility Rate: The eligibility rate was calculated as Eligibility Rate = (NE/NS) × 100, where NE is the number of eligible participants, and NS is the number of participants screened. The target eligibility rate was considered when 50% or more of the individuals screened for the study met the eligibility criteria.

Recruitment Rate: The recruitment rate was computed as Recruitment Rate = (NR/NE) × 100, where NR is the number of randomized participants, and NE is the number of eligible participants. The success of the study’s recruitment was determined when 80% of the sample enrolled in a two-month period.

Retention Rates: The retention rate was estimated as Retention Rates = (NA/NR) × 100, where NA is the number of participants assessed at T3, and NR is the number of randomized participants. Adequate participant retention was declared when more than 80% of participants were retained in the study after the post-follow-up period.

Adherence Rate: The adherence rate was calculated as Adherence Rate = (NA/NP) × 100, where NA is the number of participants who attended 80% of the sessions, and NP is the total number of participants. Program adherence was declared when 80% of the participants attended 80% of the sessions.

Acceptability: Acceptability determines how well the target population receives a program and whether it meets the needs of the target population [30]. The TAAS [31] assessed the acceptability of the DSM Coaching Program. The TAAS is a ten-item Likert scale with values ranging between 1 (*strongly disagree*) and 7 (*strongly agree*). The DSM Coaching Program was considered acceptable when 75% of the participants scored above the mean value of the scale.

Fidelity of DSM Coaching Program: The fidelity of the DSM Coaching Program was assessed through two methods: a DSM Coaching Fidelity Measure and a Comprehensive Intervention Fidelity Guide. Every individual coaching session was recorded for quality assurance purposes, and to assess the fidelity of the coaching programs. The Diabetes Self-Management Coaching Fidelity measure, adapted from the Occupational Performance Coaching Fidelity measure [32] was used to assess fidelity of coaching sessions. The checklist was adapted by the research team (FY, CD, and DK) and two occupational therapists (YA and RG) working in the study setting through an iterative review process. The DSM Coaching Fidelity Measure has 23 items categorized into three parts: critical element, client response, and distinguished items. The first or the initial sessions had 16 items, and the subsequent sessions had a total of 23 items. Overall, the session was categorized into initial sessions with 16 items, and subsequent sessions encompassed 23 items, including the initial session items. The fidelity of the coaching sessions was calculated for each week and computed to the fidelity of the overall session. Hence, the fidelity of the coaching session was determined as satisfactory when the fidelity score of the overall session was above eighty percent. On the other hand, the fidelity of the overall DSM Coaching Program was examined using the Comprehensive Intervention Fidelity Guide [33]. The guide had four components: design, intervention, delivery, and recipient. The guide had 22 items rated from 0 to 2, where 0 = absent, 1 = moderate, and 2 = extensive fidelity. A fidelity score of 22 is moderate, and 44 is the highest, representing extensive fidelity. The fidelity of the DSM Coaching Program was assessed by a research assistant who was familiar with the overall intervention process.

### 2.6. Data Analysis

Data pertaining to the program’s feasibility, acceptability, and fidelity were entered to Epi Info version 7.2, imported to an excel spreadsheet, and then exported to Statistical Package for Social Sciences version 29 for further analysis. Descriptive statistics were computed to determine the frequencies, percentages, means, medians, modes, standard deviations, and interquartile ranges. We computed the Shapiro–Wilk test [34] and Whisker plots to examine the normality assumption and presence of an outlier, respectively. Normally distributed data were analyzed using a parametric test: the independent sample Student’s t-test was used to detect the mean differences between the treatment and control groups at baseline. For data not normally distributed we conducted non-parametric tests, such as the Mann–Whitney U test, to determine the baseline mean difference between the groups. We used the chi-square test and Fisher’s exact test for categorical variables in respect to the assumptions. A *p*-value < 0.05 with a 95% Confidence Interval (CI) was taken as a statistically significant association.

The inductive qualitative content analysis technique was used to analyze the qualitative data [35,36]. All of the qualitative interview records were transcribed verbatim into Amharic by an experienced transcriber. The principal investigator read the transcripts repeatedly to understand the essence and validate the transcription. Two transcripts were selected and translated into English by a certified bilingual language translator. The research team (FY, CD, and DK) coded the two translated transcripts individually to prepare a codebook. The research team sat together to discuss codes, resolved discrepancies through discussion, and prepared a codebook. The final draft of the codebook was reviewed, commented on, and modified by the research team before approval, and the team agreed to discuss emerging codes. The codebook was translated into Amharic so that the rest of the interviews could be coded. Finally, the codes were merged into categories and then into themes.

**Study protocol**: The protocol of the study was reviewed and published by a peer-reviewed journal [37].

## 3. Results

### 3.1. The Sociodemographic Characteristics of the Study Participants

Forty participants, twenty in each group, were recruited for the study. There was no baseline statistical difference between the treatment and control groups. Many of the participants in the treatment and control groups were females, married, and Orthodox Christians. About 35% of both groups of participants had been living with diabetes for more than five years. About two-thirds of the participants in the treatment and control groups had one or more comorbidities. Most of the participants in the treatment (70%) and control (65%) group were taking oral hypoglycemic agents (Table 1).

### 3.2. Feasibility and Acceptability of the DSM Coaching Program Eligibility Rate

A total of 252 individuals with type 2 diabetes were assessed for eligibility for the DSM Coaching Program (Figure 1). The most frequently noted exclusion criteria were HbA1c ≤ 7 (30.6%) and diabetes complications (29.4%) (Figure 2).

Recruitment Rate: Out of 57 eligible participants, the study enrolled 40 (70%) participants in one month (Figure 1).

Retention Rates: The retention rates of the DSM Coaching Program at post-intervention and post-follow-up were 92.5% and 90%, respectively, which are slightly higher than the target retention rate of 80%.

Adherence Rate: The mean session attendance of the DSM Coaching Program was 8.15 ± 2.4 (95% CI: 7.01–9.29). Seventeen participants attended eight or more sessions. As a result, the adherence rate of the DSM Coaching Program was 85%, which is a bit higher than the target adherence of 80% (Figure 2).

The Acceptability of the Program: More than sixty percent of the participants responded positively to each TAAS item. The mean TAAS assessment score was 6.7 ± 0.24 (95% CI: 6.61–6.85), and it shows the overall acceptability of the program.

To triangulate the above feasibility and acceptability findings, ten participants from the treatment group were interviewed to explore their perspectives about the DSM Coaching Program. Of these participants, 50 percent were females and had less than secondary education. As a result, four themes were generated from the qualitative data analysis: Theme 1: the coaching program is life-changing, Theme 2: personal and contextual factors, Theme 3: coaching enhances commitment, and Theme 4: the perceived benefits and outcomes of the program.


*Theme 1: The Coaching Program is Life-Changing*


All of the participants expressed their satisfaction with the program and its benefits. They described the relevance of the coaching program. More than half of the participants mentioned that the DSM Coaching Program components (diet, exercise, medication, blood test, and foot care) were relevant and brought about change in their lives. For example, a female participant described her feelings as follows: “For me, the Coaching Program was a lifesaver” (P20).

The majority of participants were pleased with the coaching approaches (group and individual coaching). The group coaching program facilitated experience sharing among participants. For example, a female participant mentioned the following: “Both the group and individual home-based coaching sessions worked for me. They [sessions] allowed us [participants] to discuss freely with the coach as a friend” (P03).

Most respondents suggested that the DSM Coaching Program should be continued and expanded to reach a wider diabetic community. A male participant described the relevance of the program as follows: “I think it is nice to widen the scope of the training; for example, we [participants] get this chance because we are living in the city.” (P39).

The participants also highlighted the potential acceptability of the DSM Coaching Program by other people living with diabetes. One male participant said “Others [people with diabetes] will accept the coaching program. It is 100% acceptable because the program is vital.” (P11).


*Theme 2: Personal and Contextual Factors. “Why did you come without an appointment?”*


This theme summarizes personal-, social-, and system-level challenges and enablers of participation and the implementation of the diabetes self-management principles.

The flexibility and convenience of the coaching program, involvement of family members, and reimbursement of transport costs enhanced participation. Most of the participants did not identify any personal challenges that affected their participation. One female participant said, “I faced no problems; I had never experienced any challenge during the training. I was eagerly waiting for the training day” (P03). Another male participant affirmed that “There was no financial or social problem that affected my participation in the program because the program was in a flexible schedule at the nearest health center” (P14).

Many participants mentioned that participating in DSM coaching did not affect their social roles. However, some participants pointed out that religious practices, household responsibilities, and social events like weddings and funerals interfered with participation. A male participant explained, “We might be busy with funerals, weddings, religious events, or other programs to perform regular exercise” (P04).

The health system is another potential challenge for the successful implementation of self-management activities, with issues related to the continuity of services. The participants highlighted that the interventions within the DSM Coaching Program are not something they would have regularly received in the current health system. A female participant revealed,
I have never been given education at the hospital; you may be surprised. Though I have been monitored by doctors at the hospital, I don’t have a regular doctor to follow me. If I meet one doctor today, I will meet another on my next visit. Further, they don’t tell you anything other than the increase or decrease of your sugar level.(P20)

One male participant shared his experience in the clinic: “If you show up without your appointment, they will not be happy and will be annoyed and ask you, ‘Why [did] you come without an appointment?’” (P11).

While the current system is not structured to provide interventions included in the DSM Coaching Program, the participants felt that this type of program should be more accessible to individuals with type 2 diabetes. One participant noted, “The program was crucial; the coaching program should continue in the health centers and at home. If the program continued, many people would benefit from it” (P14).


*Theme 3: Coaching Enhances Commitment*


The DSM Coaching Program enhanced the commitment of the participants to carry out self-management activities, highlighting their motivation, increased confidence, and the important influence of the coach.

The study participants mentioned that the DSM Coaching Program increased their motivation to practice self-management. Most participants were committed to applying self-care activities; for example, a female participant noted, “I am committed to continue applying the self-management activities as I saw hope in it. I will pray to God to give me strength” (P34).

The participants highlighted that the DSM Coaching Program boosted their self-confidence in performing diabetes self-management activities regularly. A female participant noted, “My self-confidence to apply the self-management activities has just increased. I have no hesitation in applying the recommendations; that is what I told myself” (P34).

All participants mentioned that the coach was effective and contributed to their commitment. A female participant said “It’s hard for me to describe the coach; he was just like a brother. He taught me basic knowledge and skills about diabetes management” (P34).


*Theme 4: The Perceived Benefits and Outcomes of the DSM Coaching Program*


This theme summarizes the benefits of the program, the changes in health status, and participants’ satisfaction. Some participants described their changes to health behaviors and the resultant improvements in their health statuses. For example, a male government employee participant explained his behavior change: “Now I left all the malpractice behind and became very careful. I get a lot of improvement and changes in my life” (P14).

In this study, the participants reflected that involving family members in the training created a better understanding of the illness and support for its management. One participant mentioned, “After the training, my relationship with my family has improved” (P32). Another woman said “The program created an understanding for my family to give attention and support. Now my family has learned what to do and started to tell me what to eat” (P34).

### 3.3. Data Integration

To address the feasibility and acceptability of the DSM Coaching Program objectives and fully understand the essence of the process, the quantitative and qualitative results were presented in a joint display technique (Table 2). The findings from both strands show congruence and explain the feasibility and acceptability of the DSM Coaching Program.

### 3.4. The Fidelity of the DSM Coaching Program

Seventy-one individual coaching sessions were recorded for the fidelity assessment. The fidelity of both the individual coaching sessions and the overall coaching program was assessed using a structured checklist. An occupational therapist assessed the fidelity of the recorded coaching sessions using the Diabetes Self-Management Coaching Fidelity Measure adapted from the Occupational Performance Coaching Fidelity Measure [32]. The initial session had 16 items, and the total score was 48. The fidelity score of the initial coaching sessions (week 4) was 45 (93.7%) out of 48. The fidelity score of the subsequent coaching sessions at weeks 6, 8, and 10 was computed from the total item score of 69. Accordingly, the fidelity score of the three subsequent sessions at weeks 6, 8, and 10 were 94%, 96.8%, and 94.5%, respectively. Hence, the overall fidelity of the coaching sessions was 94.7%, which is far above the benchmark of 80%. In addition, the fidelity of the DSM Coaching Program was assessed by a research assistant who participated in the study processes. The fidelity assessment was conducted using the Comprehensive Intervention Fidelity Guide. The guide had 22 items, and the total score for each item was 44. The fidelity score of the DSM Coaching Program was 37 (88.6%) out of 44, which is above the moderate level.

## 4. Discussion

This study aimed to determine the feasibility, acceptability, and fidelity of the DSM Coaching Program among individuals with type 2 diabetes in primary care in Ethiopia. It is the first study to introduce health coaching in this context. The concurrent single-blinded, parallel group mixed-method RCT study proved the feasibility, acceptability, and fidelity of the DSM Coaching Program for individuals with type 2 diabetes in this context. This study reported excellent recruitment, adherence, retention, and acceptability rates.

Recruiting adequate study participants is a critical step to ensure the quality of a study, especially in clinical trials, and failure to recruit sufficient participants could lead to failure to meet the study objectives [38,39]. While the recruitment plan was to enroll 90% of the participants in two months, this study met the target recruitment of 40 participants in just one month, giving a 100% recruitment rate. This achievement was due to the use of audio–visual recruitment information in the waiting area and the involvement of all of the nurses working in the diabetes clinic in the screening process. A study by Iben and colleagues supported this finding and suggested that audio-visual materials enhanced participant recruitment [38]. The recruitment rate of the current study is higher than that of a feasibility study examining a structured self-management education program for type 2 diabetes in the primary care of LMICs, Malawi, and Mozambique, which reported 81.6% [40]. This difference could be due to the exclusion criteria. Unlike the current study, the study in Malawi and Mozambique excluded participants who did not want group participation. In addition, the recruitment rate of the current study is significantly higher than those of studies in the United States of America (35.2%) [41], Canada (29%) [42], and the UK (26.2%) [43]. This variation might be due to a difference in the eligibility criteria, recruitment period, and recruitment strategy [44]. Moreover, the higher demand for health education in this setting, awareness created before recruitment, and program accessibility for the participants could be the reasons for better recruitment in the present study. The qualitative findings of this study highlight the lack of health education services in this setting. On the other hand, health facilities are the primary source of information for individuals with diabetes in Ethiopia [45].

Achieving the maximum retention of study participants improved the efficiency of the study and minimized study biases [46]. The current study successfully retained the majority of the participants and met the target retention rate after three months of follow-up. The reasons for the successful retention of participants in the current study were the higher demand for health education in the study area, the perceived effectiveness of the program, the involvement of family members in the program, and the accessibility of the program to the participants. The qualitative findings of the current study also support this. In addition, this finding is supported by a study by Poongothai and colleagues, who studied strategies for participant retention in long-term clinical trials. They found that building effective communication between the research team and study participants leads to successful participant retention [47].

The retention rate of the study is almost consistent with a survey conducted in the United States of America among Mexican patients with type 2 diabetes [48]. However, the retention rate of this study is lower than that of a feasibility study by Whitehouse and colleagues, which demonstrated 100% retention [49], and a study in Malawi and Mozambique, which had a 94% retention rate [40]. This difference could be due to the differences in the number of sessions, criteria for success in retention, and the duration of intervention delivery. Evidence shows that studies with a longer intervention duration demonstrate lower retention rates [50].

On the other hand, the retention rate of the current study is higher than that of a study conducted among church attendants in the Republic of Marshal Island, which retained about 56% of participants [51]. This difference might be due to a variation in population; participants in the faith-based study might be healthier and more stable, so they may have withdrawn from the study [52]. In addition, the retention rate of the current study is higher than that of studies conducted in Canada, at 80% [47], and Denmark, at 70% [53] (1–1). This difference might be due to the variation in the coaching approach employed. Unlike the studies in Canada and Denmark, the current study used a hybrid approach (group and one-on-one coaching), enhancing participant engagement and retention. It is evidenced that group-based interventions enhance participant retention [50,54].

The adherence rate of the DSM Coaching Program is slightly higher than the target adherence rate. The program’s flexibility, compensation for missed sessions, participatory session planning, the convenience and accessibility of the coaching venues, the involvement of family members, the relevance of the program, and reimbursement of transport costs were strategies applied to achieve the target adherence. These strategies are supported by different studies conducted on intervention adherence [55,56].

The context of primary care in Ethiopia is very conducive to integrating self-management programs with the routine health extension program and enhancing adherence to the intervention because health extension workers spend 75% of their time in household visits and outreach services and 25% at the health post [57], which could be ideal to provide behavioral change interventions for individuals with type 2 diabetes at the community level. In light of the above evidence, a study conducted in Gondar town to assess the utilization of urban health extension workers revealed that about 60% of households utilized services by health extension workers [58]. Unlike the rural health extension package, the urban health extension program includes non-communicable diseases and is led by diploma nurses who have basic knowledge and skills in non-communicable disease prevention, management, and complication prevention [57].

Micro-, mezzo-, and macro-level factors affect adherence to participating in and implementing the DSM Coaching Program. The qualitative strand of the current study revealed that personal, socio-cultural, and health system factors affect participation and the implementation of the DSM Coaching Program. Most of the respondents affirmed that there was no personal barrier that affected their participation and the implementation of the DSM Coaching Program.

The participants also stated that the health system is another challenge to implementing the DSM Coaching Program. The current way in which they receive diabetes support within the health system is largely through specialized outpatient clinics, which are too crowded to obtain quality healthcare. Nonetheless, diabetes is a lifelong health condition that requires collaborative and continuous support from family, society, and the health system. Given primary care’s focus on continuous care across the lifespan and community participation, it is an ideal setting to provide self-management support. The current structure of the health system in Ethiopia is not prepared for the existing chronic disease burden and self-management demand of individuals with chronic diseases. However, Ethiopia has a well-developed health extension program that would be ideal for leveraging diabetic self-management support programs. The qualitative findings of the current study also show that the coaching program was an opportunity to learn about diabetes self-management because structured diabetic health education is seldom given in diabetic clinics in Ethiopia [59]. Gudina et al. found that there are no diabetic educators and diabetic educator training programs in the country.

The TAAS conducted in the current study demonstrated an excellent acceptability of the coaching program at the item and scale levels. All participants assessed with the TAAS demonstrated acceptance of the DSM Coaching Program. This finding is congruent with the results of the qualitative section of this study. Most participants were satisfied, benefited from the program, and perceived that the DSM Coaching Program would be accepted by other individuals with type 2 diabetes. This might be due to the implementation of an evidence-based DSM Coaching Program that is culturally and contextually adapted to the local context by a locally available multidisciplinary team.

The current study’s acceptability findings are consistent with a pilot study on older adults with type 2 diabetes in Thailand [55]. Another process evaluation study on medical assistance health coaching in the United States of America supported the acceptability of health coaching programs among individuals with type 2 diabetes [60]. Furthermore, the current study’s findings are concurrent with those of a qualitative study on self-management support programs among individuals with type 2 diabetes in Canada [61].

Ensuring the fidelity of behavioral change programs is a prerequisite to achieving the effectiveness of an intervention [62]. The overall fidelity of the coaching sessions and the DSM Coaching Program was excellent and met the target fidelity benchmark, giving a green light to the future definitive trial. The fidelity finding is consistent with a diabetes prevention study that demonstrates acceptable fidelity [63]. Furthermore, this finding is supported by a phone-based coaching program that examined the feasibility and acceptability of the program among patients with diabetes [64].

### The Strengths and Limitations of the Study

One of this study’s strengths was that it adapted an evidence-based intervention using an iterative process by a multidisciplinary team. Employing a mixed-methods RCT design to triangulate the feasibility outcomes was of added value. This study also assessed the fidelity of the DSM Coaching Program, indicating that the program is implementable in the future. However, the fidelity measure and the TAAS checklist have not been validated; therefore, the fidelity findings must be interpreted with caution. The study did not address healthcare providers’ perspectives and policy evaluation on the program’s feasibility and acceptability. The small sample size and the fact that it was a single-center study might limit the generalizability of the study.

## 5. Conclusions

The current study showed remarkable eligibility, recruitment, retention, adherence, and acceptability rates, which supports the planning of a definitive effectiveness trial. As a result, it is feasible to plan and implement the DSM Coaching Program in the primary care context of Ethiopia. The DSM Coaching Program was found acceptable by individuals with type 2 diabetes. The program ensured acceptable fidelity in implementing the DSM Coaching Program. The study findings will provide a clear direction for implementing future definitive trials in primary care settings in resource-limited settings.

## Figures and Tables

**Figure 1 ijerph-21-01032-f001:**
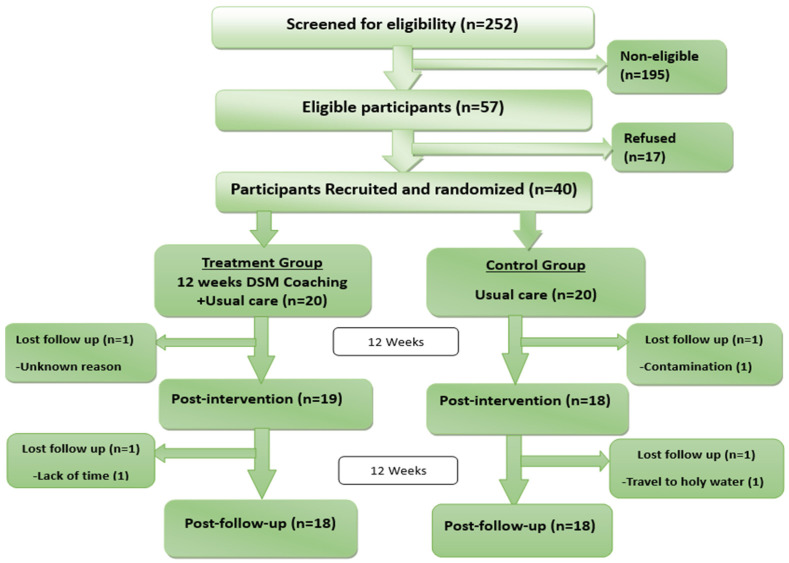
A consort flow diagram showing the eligibility, recruitment, and retention rates of the study participants. The boxes on the right and left represent the control and treatment group participants, respectively and ‘n’ denotes number of participants.

**Figure 2 ijerph-21-01032-f002:**
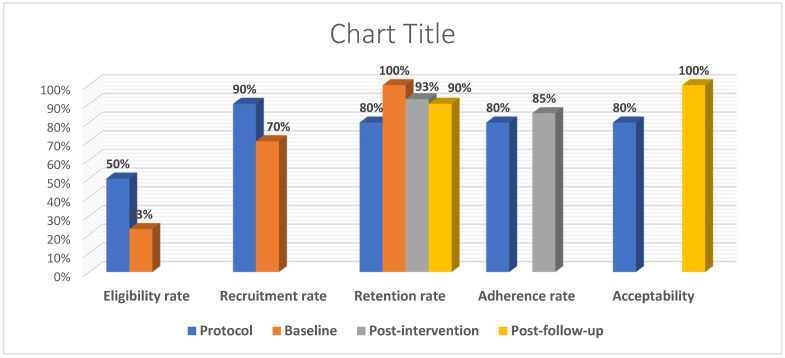
A summary of the feasibility outcomes of the study at the baseline, post-intervention, and post-follow-up periods compared to the target in the protocol.

**Table 1 ijerph-21-01032-t001:** Baseline sociodemographic and clinical characteristics of the DSM Coaching Program study participants in Ethiopia, 2023.

Variables	Treatment n (%/IQR)	Control n (%/IQR)	*p*-Value
**Age ****	55.5 (10) **	55 (23) **	0.75
**Age at Diagnosis ****	48 (12)	50 (13)	0.8
**Sex**			
Male	6 (30)	3 (15)	0.46
Female	14 (70)	17 (85)	
**Education**			
Unable to Read and Write	3 (15)	6 (30)	0.42
Elementary (1–8)	8 (40)	3 (15)	
High School (9–12)	6 (30)	5 (25)	
College and University	3 (15)	6 (30)	
**Religion**			
Orthodox Christian	17 (85)	16 (80)	0.37
Muslim	2 (10)	4 (20)	
Protestant	1 (5)	0 (0)	
**Occupational Status**			
Private Employee	1 (5)	1 (5)	1.0
Government Employee	3 (15)	3 (15)	
Housewife	8 (40)	8 (40)	
Merchant	5 (25)	5 (25)	
Other	3 (15)	3 (15)	
**Marital Status**			
Married	12 (60)	14 (70)	0.70
Divorced	3 (15)	1 (5)	
Widowed	5 (25)	5 (25)	
**Budget for Diabetes Care**			
Health Insurance	13 (65)	15 (75)	0.28
Out of Pocket	7 (35)	3 (15)	
Free Health Coverage	0 (0)	2 (10)	
**Duration Of DM**			
<2 years	4 (20)	3 (15)	0.67
2–5 years	9 (45)	10 (50)	
>5 years	7 (35)	7 (35)	
**Comorbidity**			
No	7 (35)	7 (35)	1.00
Yes	13 (65)	13 (65)	
**Number of Comorbidities**			
One	1 (5)	1 (5)	0.29
Two	12 (60)	9 (45)	
Three	0 (0)	1 (5)	
**Type of DM Medication**			
Insulin injection	2 (10)	1 (5)	0.46
OHA/s	14 (70)	13 (65)	
Both	4 (20)	6 (30)	
**Hba1c ****	7.75 (3.2)	7.6 (2.8)	0.69
**Systolic Blood Pressure ***	119 ± 14.7	124 ± 17.2	0.4
**Diastolic Blood Pressure ****	70 (10)	80 (10)	0.23
**Body mass Index**	26.96 ± 3.57	25.2 ± 4.2	0.16

Key: *—Mean; **—Median; n—Frequency.

**Table 2 ijerph-21-01032-t002:** A side-by-side presentation of the qualitative and quantitative feasibility findings.

Quantitative Feasibility Outcomes	Themes and Sub-Themes of the Qualitative Findings
**Recruitment:** This study achieved a 100% recruitment rate in half the recruitment period. The majority of participants believe that the program would effectively manage diabetes (TAAS report).	Theme 1. The coaching program is life changing. P03: “They (researchers) can announce in the waiting room and register volunteers as they recruit me and other participants. That means people can register and attend the training.”
**Retention:** The program retained 90% of the participants at the end of the study. About 95% of participants are committed to completing the program if they start (TAAS report).	Theme 2. The flexibility and convenience of the program. P14: “The program was scheduled based on the availability and convenience of participants and was flexible.” Theme 3. The coach was an effective educator. P03: “The coach is very good. He teaches us very politely and effectively.” Theme 3: Coaching enhances commitment. P04: “The group discussion keeps me engaged, sharing experiences, and learning from others’ experiences.”
**Adherence to the program:** More than 85% of the participants reported compliance with the program. The TAAS report also shows that over 94% of the respondents were committed and adherent to finish the program once they started.	Theme 1. The coaching program is life changing. P34: “I didn’t miss any part of the coaching program because I benefited from it.”
**Acceptability:** All participants agreed that they would not drop the program if they started it.	Theme 1. The coaching program is life changing. P16: “Yes, people will accept the program. If you want to teach patients about their problem.”

## Data Availability

All required data will be available upon request to the principal investigator.

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
