# Peer review of "The Feasibility of the Diabetes Self-Management Coaching Program in Primary Care: A Mixed-Methods Randomized Controlled Feasibility Trial"

_ijerph, 2024, doi:10.3390/ijerph21081032_

Round 1

Reviewer 1 Report

Comments and Suggestions for Authors

The paper titled "Feasibility of the Diabetes Self-Management Coaching Program in Primary Care in Ethiopia: A Mixed Method Randomized Controlled Feasibility Trial" is carefully read and reviewed. The authors studied the feasibility, acceptability, and fidelity of a Diabetes Self-Management Coaching Program in primary care in Ethiopia, given the absence of such support strategies in the country. Authors designed a single-blinded, parallel-group, mixed-methods feasibility randomized control trial. The adult patients with HbA1c levels ≥7%, who were on diabetic medication and residing in Gondar town, were included. The participants were allocated into treatment and control groups using a block randomization technique with a block size of four. The treatment group received a 12-week Diabetes Self-Management Coaching Program in addition to usual care, while the control group received only the usual care. Authors concluded that the study demonstrates the Diabetes Self-Management Coaching Program is feasible, acceptable, and implementable in the primary care setting in Ethiopia.

There are several concerns about the study, which should be revised; 

1-Title is concise but abstract does not summarize the text adequately, especially, methodology.

2- Background data, study rationale and objectives are clear in introduction.  Methods were expressed comprehensively in the text and statistics are also accurate. However, normality analysis was not mentioned. In addition, power analysis would be better to calculate the sample size of the study.

3- Results section is too long and difficult to follow the study results. Maybe authors can consider additional figures to make following of the results easier.

4- Since the results of the study are specific to a particular region of the world, I believe it would be more appropriate to consider the article for a more localized journal.

5- There are improper self citations. 

Author Response

Comment 1- Title is concise but abstract does not summarize the text adequately, especially, methodology.

Response 1- Comment accepted. Thanks for the raising this issue. Methodology section of the abstract has been revised. We revised the statistics part of the method and added the following sentences “Descriptive statistics such as frequency, mean, median, and standard deviations were computed. Based on the normality assessment, baseline group difference was examined using independent sample student t-test, Mann-Whitney U test, and chi-square test”. (See the abstract section of the revised manuscript page 2 line 32-42).

Comment 2- Background data, study rationale and objectives are clear in introduction.  Methods were expressed comprehensively in the text and statistics are also accurate. However, normality analysis was not mentioned. In addition, power analysis would be better to calculate the sample size of the study.

Response 2- Comment regarding normality assessment has been accepted.

  1. Normality test: We did not include inferential statistics in this manuscript; however, we provided information on the normality assessment of the baseline variables. “Descriptive statistics were computed to determine Frequencies, percentages, means, medians, modes, standard deviations, and inter quartile ranges were computed to describe the data. To make a baseline comparison, normality assumption and outlier assessment were checked using a Shapiro-Wilk test and Whisker plots, respectively. Normally distributed data were analyzed using a parametric test. We used a independent sample student t-test to detect mean differences between the treatment and control groups. We conducted non-parametric tests for data that failed to fulfill the normality assumptions. We applied a Mann-Whitney U test to determine the mean difference between the groups. For categorical variables chi-square test and Fishers exact test were computed depending on the chi-square assumptions. P-value < 0.05 at a 95% Confidence Interval (CI) was taken as a statistically significant association”. (please see the data analysis section of the method on page 12 line 256-266).
  2. Power analysis for sample size calculation. We appreciate your insightful suggestion. Yes, power analysis is the best way to estimate sample size specially for definitive trials. For pilot/feasibility studies that are designed to provide data to inform fully powered studies, it is suggested to follow rule of thumb to estimate sample size. Several researchers suggest different methods of estimating sample size for pilot study. We follow Melody A. Herzog’s suggestion of 10-40 per arm as a sample size. We acknowledge the different techniques of sample size calculation for pilot/feasibility studies.  

Comment 3- Results section is too long and difficult to follow the study results. Maybe authors can consider additional figures to make following of the results easier.

Response 3-. Thank you for this feedback.  Because of the mixed methods approach we attempted to reflect on the three study objectives the result section has become long. To make the result section more concise and clear we: 1. revised the result section and removed repeated quotes and additional explanations, 2.  merged the two baseline tables in to one, and 3. modified table 2 and gave some details within it, and 4. added a figure that summarise the feasibility outcomes.

Comment 4- Since the results of the study are specific to a particular region of the world, I believe it would be more appropriate to consider the article for a more localized journal.

Response 4- We appreciate the suggestion; however, we believe the study could have a significant contribution in the knowledge and evidence generated in the arena of diabetes self-management and diabetes health coaching. As diabetes mellitus is a global health problem affecting more than half a billion people, to tackle the existing and impending burden of the disease, the clinical and scientific community could benefit from diverse perspectives and contexts. Moreover, health coaching is an emerging self-management support little is known in low- and middle-income countries, particularly sub-Saharan regions, despite the fact that people in these regions are the primary victims of an increased diabetes related mortality, morbidity, and disability. The evidence generated from this study might help not only the local community but also the sub-Saharan regions and other low-income settings. The current study introduced a hybrid approach (group + individual home-based coaching) that showed a promising effect on reducing blood sugar. Hence, this could be a pertinent lesson for the global community working on diabetes.  

Comment 5- There are improper self citations. 

Response 5- We have reduced the number of self- citations.

Reviewer 2 Report

Comments and Suggestions for Authors

Comments to the Authors,

I read with great interest the manuscript ID ijerph-3062822 entitled " Feasibility of the Diabetes Self-Management Coaching Program in Primary Care in Ethiopia: A Mixed Method Randomized Controlled Feasibility Trial".,   The manuscript tackles an important issue. However, it has some methodological pitfalls.

Title:

It would be nice to specify the group assessed i.e people with T2DM. It’s a single center study, it would be nice to clarify that.

Aim:

I see the specific objectives are repetition of the objectives, it’s better to omit them.

Methodology:

·      It would be nice to add some comparisons between the two studied arms before and after the study period.

·      For how long was the study done, how about the drop out rate?

·      The study depends on a questionnaires, was this questionnaires validated. Moreover, the language is in Amharic, was the translation of the questionnaire tested and validated? 

·      The sample size of the study is too small and it represents single center rendering the data not representative, it would be nice to add this to the limitations.

Results:

It would be nice to add comparison of both groups before and after the program. 

Comments on the Quality of English Language

Author Response

Comment 1-Title: It would be nice to specify the group assessed i.e people with T2DM. It’s a single center study, it would be nice to clarify that.

Response 1- Thank you for pointing out this issue. We added individuals with type 2 diabetes and a single center in the title. “Feasibility of the Diabetes Self-Management Coaching Program Among individuals with Type 2 Diabetes in Primary Care in Ethiopia: A Single Center Mixed Method Randomized Controlled Feasibility Trial” (See the running title of the manuscript)

Comment 2- Aim:

I see the specific objectives are repetition of the objectives, it’s better to omit them.

Response 2- We appreciate the comment and have removed repetitive words from each statement. We do believe the specific objectives in the initial version addressed feasibility, acceptability, and fidelity as each has their own outcome measurement tools and are stand-alone outcomes.  For clarity and understanding we have continued to keep these specific objectives.

Comment 3- Methodology: It would be nice to add some comparisons between the two studied arms before and after the study period.

Response 3- Because this study is a feasibility study and the objectives are to examine feasibility outcomes, we have not included outcome data. The study has both quantitative and qualitative strands to address the feasibility outcomes and both provide adequate detail of evidence to answer the feasibility, acceptability, and fidelity questions.

Comment 4- For how long was the study done, how about the dropout rate?

Response 5- Thanks for the comment. The study period was mentioned in the study setting section and was conducted from November 1, 2022, to April 30, 2023. The drop out rate was described in the result section under retention rate. The attrition rate or drop out rate is the inverse of retention rate. The retention rate of the study at the post-intervention and post-follow-up was 92.5% and 90%, respectively. Meaning the drop out rate is 7.5% and 10% respectively.  This data has been included in the manuscript.

Comment 6- The study depends on a questionnaire, was this questionnaire validated. Moreover, the language is in Amharic, was the translation of the questionnaire tested and validated?

Response 6- We appreciate raising this issue and have provided further clarification and acknowledged this limitation. This manuscript used two checklists, the fidelity assessment and the intervention acceptability checklists. These checklists did not undergo formal validity and reliability assessment. However, the checklists were culturally adapted and translated to local language, Amharic, by language experts and health care professionals before use. We have included this under the limitation section of the study. “However, the fidelity measure and the TAAS checklist has not been validated and therefore fidelity findings must be interpreted with caution”. (Please see the strength and limitation section on page 28 line 531-533).

Comment 7- The sample size of the study is too small, and it represents single center rendering the data not representative, it would be nice to add this to the limitations.

Response 7- This study is a pilot/feasibility study and the sample size is appropriate for this design as per expert’s suggestion. The main purpose of this study is to examine the feasibility and acceptability of the DSM Coaching program by individuals with type 2 diabetes in the primary care setting. Unlike the definitive trial, this study doesn’t need to be representative and generalizable to the larger diabetic community. As it is a feasibility study and its intention is to see whether the main definitive trial is possible or not, the suggested sample size is enough to address the feasibility outcomes.

Comment 8- Results: It would be nice to add a comparison of both groups before and after the program. 

Response 8- We appreciate the comment, however, the purpose of this study is to address feasibility outcomes and not to compare outcome difference between the two groups. We have conducted baseline comparison and presented this in Table one, and also provided detailed evidence on the feasibility, acceptability, and fidelity of the DSM Coaching program.

Reviewer 3 Report

Comments and Suggestions for Authors

Lines 203 – 299 – it is necessary to remove the general definitions of the mentioned terms. Leave the calculation of the number of respondents and the percentage related to this pilot study.

What statistical methods were used to test the significance of differences between individual variables? Unclear and based on the sample size and assumed data distribution it looks like the wrong statistical methods were used (eg Student's T-test, Chi-square test for less than 5 subjects?).

Results

The "s" in the title is missing - it says “result”

Table 1. The median cannot be shown as mean ± sd. Both values should be given in medians because the sample is tiny, and age is often a variable with irregular distribution.

Table 2. Comment for the median as in Table 1.

References are not written according to the instructions for authors.

I think that the list of abbreviations after the conclusion is unnecessary. They need to be integrated into the text of the manuscript.

Author Response

Comment 1- Lines 203 – 299 – it is necessary to remove the general definitions of the mentioned terms. Leave the calculation of the number of respondents and the percentage related to this pilot study.

Response 1- Thank you for your reflection.  We have removed the definitions for the feasibility outcome variables.

Comment 2- What statistical methods were used to test the significance of differences between individual variables? Unclear and based on the sample size and assumed data distribution it looks like the wrong statistical methods were used (e.g. Student's T-test, Chi-square test for less than 5 subjects?).

Response 2- Thank you for raising this concept. We used different statistics to compare the baseline data based on the normality distribution for the continuous variables and chi-square test for categorical variables. “Before the comparing the baseline data the normality assumption and presence of an outlier were checked using a Shapiro-Wilk test and Whisker plots, respectively. Normally distributed data were analyzed using a parametric test. We also ran an independent sample t-test to examine baseline mean difference between the groups. We conducted non-parametric tests for data that failed to fulfill the normality assumptions. We applied a Mann-Whitney U test to determine the mean difference between the groups. For categorical variables chi-square test and Fishers exact test were computed depending on the chi-square assumptions. P-value < 0.05 at a 95% Confidence Interval (CI) was taken as a statistically significant association”. This has now been included in the paper, (please see page 12, line 256-266).

Results

Comment 3- The "s" in the title is missing - it says “result”

Response 3- Thank you for noting this.  The “s” has been added to result.

Comment 4- Table 1. The median cannot be shown as mean ± sd. Both values should be given in medians because the sample is tiny, and age is often a variable with irregular distribution.

Response 4- Thank you for pointing out this issue. We revised the median along with its interquartile range and mean with its standard deviation. We also put present age at diagnosis with median based on your suggestion despite it was normally distributed. (see Table 1)

Comment 5- Table 2. Comment for the median as in Table 1.

Response 5- Thank you for noting this.  The same action has been taken on Table 2.

Comment 6- References are not written according to the instructions for authors.

Response 6- Thank you for this feedback.  The list of authors in the references have been revised using endnote citation manager. (See list of references).

Comment 7- I think that the list of abbreviations after the conclusion is unnecessary. They need to be integrated into the text of the manuscript.

Response 7-  The abbreviations listed after conclusion has been removed.

Round 2

Reviewer 1 Report

Comments and Suggestions for Authors

Revisions and authors' reply are satisfactory. I do not recommend further revision. 

Reviewer 2 Report

Comments and Suggestions for Authors

All comments were addressed sufficiently. 

Comments on the Quality of English Language

Minor editing is required.